# High Adsorption of Hazardous Cr(VI) from Water Using a Biofilter Composed of Native *Pseudomonas koreensis* on Alginate Beads

**DOI:** 10.3390/ijerph20021385

**Published:** 2023-01-12

**Authors:** Lourdes Diaz-Jimenez, Sandy Garcia-Torres, Salvador Carlos-Hernandez

**Affiliations:** Sustentabilidad de los Recursos Naturales y Energía, Centro de Investigación y de Estudios Avanzados Unidad Saltillo, Ramos Arizpe 25900, Mexico

**Keywords:** biosorption, biofilm, heavy metals, bacteria immobilization, water purification, *P. koreensis*

## Abstract

Most conventional methods to remove heavy metals from water are efficient for high concentrations, but they are expensive, produce secondary pollution, and cannot remove low concentrations. This paper proposes a biological system to remove Cr(VI) from aqueous solutions; the biofilter is composed of a native *Pseudomonas koreensis* immobilized in calcium alginate beads. Lab experiments were conducted in batch reactors, considering different operating conditions: Cr(VI) concentration, temperature, pH, and time. At 30 °C and a pH of 6.6, the immobilized bacteria achieved their optimal adsorption capacity. In the chromium adsorption system, saturation was reached at 30 h with a *q_max_* = 625 mg g^−1^. By adjusting the experimental data to the Langmuir and Freundlich models, it is suggested that *P. koreensis* forms a biofilm with a homogeneous surface where Cr(VI) is adsorbed and that the bacteria also incorporates the metal in its metabolism, leading to a multilayer adsorption. On the other hand, using Fourier transform infrared spectroscopy, it was inferred that the functional groups involved in the adsorption process were O-H and C=O, which are a part of the *P. koreensis* cell wall.

## 1. Introduction

The presence of heavy metals in drinking water is an important issue regarding human health. The properties (toxicity, persistence, accumulation, and immutability) of these pollutants cause several human diseases [1]. Chromium stands out, since it is recognized as very toxic [2]; some of the reported risks to humans by chromium include carcinogenic potential, apoptotic cell death, and altered gene expression [3,4,5]. Moreover, it has been reported that at an accumulation of 0.1 mg by g body weight, chromium can cause death [6]. Chromium could be released into the environment by some industrial processes (metallurgy, agrochemical, and petrochemical industries), and other anthropogenic activities [4,7]. In aqueous solutions, this metal could be in two ionic forms Cr(VI) and Cr(III). It is accepted that Cr(VI) is 300 times more toxic than Cr(III) due to its ability to diffuse through biological membranes [8]. For these reasons, different organizations have developed regulations about chromium in water. The maximal permissible level for chromium in water is 1 and 0.5 mg L^−1^ for agriculture and public uses, respectively. In soils, the maximum levels are 510 and 280 mg kg^−1^ for industrial and agriculture/commercial/residential uses, respectively [9]. Concentrations (µg L^−1^) of chromium in rivers and lakes around the world were determined as 388.77 (Africa), 383.93 (Asia), 13.61 (Europe), 5.42 (North America), and 903.78 (South America) [7]. In Mexico, Cr(VI) is regulated in the Official Mexican Standard NOM-147-SEMARNAT/SSA1-2004, and it is considered hazardous waste by the standard NOM-052-SEMARNAT-2005. Two decades ago, the Federal Attorney for Environmental Protection (PROFEPA, report 1995–2000) reported 61 sites contaminated by heavy metals in Mexico. More recently, Cr(VI) has been found in several regions in the country [9,10,11], and more than 7300 companies involving the use of Cr(VI) were also identified [12]. These could become a source of risk for water pollution.

There are various processes for removing heavy metals from water, such as chemical precipitation, ion exchange, membrane filtration, flocculation, electrodialysis, carbon nanotechnology, and adsorption [13,14]. However, most are expensive and inefficient for removing contaminants in concentrations less than 100 ppm. In addition, these processes generate secondary waste, which represents a negative environmental impact and the necessity of additional investment to dispose of those waste materials [13,15]. Concerning chromium removal, from the early years of this century, coagulation–precipitation and adsorption have been identified as the most effective methods, but sludge disposal is their main drawback [2]. Electrochemical methods, membrane filtration, and ion exchange have also been evaluated; in these methods, technical and economic issues should be overcome before industrial-scale application [16,17]. Recently, nanomaterials have been studied in chromium removal from water; the interference with other contaminants and costs are some of the challenges of this technology [16,18].

Nowadays, biosorption is identified as another promising alternative for the removal of heavy metals at low concentrations (<100 ppm), including chromium [19]. This method requires low operation costs, is easy to implement, and generates few hazardous secondary contaminants [20]. Biosorption refers to the uptake of pollutants by either live or dead biomass through physicochemical mechanisms such as adsorption, ion exchange, redox reactions, chelation, and precipitation, among others [21]; these phenomena depend on both the type of biomass and the contaminant to be removed. Bacteria, fungi, and algae are some microorganisms studied for heavy metal biosorption [22,23]. Bacteria achieve acclimatization, adaptation, and resistance mechanisms in facing heavy metals at the cellular population and community levels [24]. In particular, the genus *Pseudomonas* has shown promising results for heavy metal removal [22,25,26].

Some drawbacks of heavy metals removal from water by bacteria are the small particle size (<5 µm) of the microbial aggregates, low mechanical resistance, and difficulty separating biomass from the effluent. It has been determined that the immobilization of microorganisms in particles of inert materials minimizes such drawbacks [27]. Several materials have been used for this purpose, including zeolites, clays, ceramic materials, alginate, chitosan, plant fibers, seeds, mycelium, and many other biomass-based materials [28,29]. However, it is still necessary to better understand biofilm formation and its interaction with heavy metals.

Considering the problem described before, in the present study, a native bacteria (*P. koreensis*) isolated from water bodies in a mining area is studied in a biosorption system for chromium removal. At this stage, lab-scale experiments were performed to identify the feasibility of *P. koreensis* in removing chromium. In this study, alginate beds were synthesized and used as microorganism carriers to minimize the inconvenience of particle size and mechanical resistance of the bacteria. After that, several operating conditions on batch reactors were tested to evaluate the capacity of the immobilized bacteria to remove Cr(VI). The experimental data were employed to determine the maximum adsorption capacity (*q_max_*) of Cr(VI) using the Langmuir and Freundlich adsorption isotherm models. In addition, pseudo-first and pseudo-second order models were considered to identify the kinetics involved in the adsorption of Cr on the *P. koreensis*/alginate beads system. It is important to remark that the presence of other elements in water (anions, cations, other metals, etc.) promotes competition for the active sites in the biofilter; this affects the selectivity and then the performance of bacteria to the sorption of chromium. Therefore, in this study, other elements were not considered to identify the efficiency of *P. koreensis* in removing chromium without interferences; this helps generate reference information that could be used in further studies. However, the long-term objective of the research is to develop a biofilter to remove heavy metals from drinking water; in the case of Mexico, this corresponds to underground water. To complement the study, analytical techniques, such as MEB, SEM, EDS, FTIR, were employed to explain the remotion of Cr(VI) by immobilized *P. koreensis* on alginate beads.

## 2. Materials and Methods

### 2.1. Biofilter Preparation

Alginate beads of different sizes were synthesized by coprecipitation, following the method reported elsewhere [30]. An alginate solution (15 g L^−1^) was placed in a separatory funnel and dropped into a calcium chloride solution (15 g L^−1^) under constant magnetic stirring. The obtained beads were washed three times with distilled water.

For the preparation of different sizes of alginate beads, the separatory funnel was conditioned by placing a micropipette tip of different capacity at the outlet, namely: blue tip (100 to 1000 μL), yellow tip (20 to 200 μL), and white tip (2 to 20 μL).

The quantification of the synthesized beads was obtained by manual count, considering three samples of 1 g.

To select the best size of alginate beads, significant differences were determined by an ANOVA at 95% confidence and 10% error. Since it is expected to obtain a high quantity of beads in each set, it is required to select a representative number of them to be measured and weighed as required by the statistical analysis. This number is computed using Equation (1).
(1)$$n=\frac{NpqK^{2}}{K^{2}pq+\left(N-1\right)E^{2}}$$
where *n* is the number of beads that must be measured to obtain a representative sample, *N* is the number of beads per set, *p* is the proportion of beads that have the characteristics of bacteria immobilization, and *q* is the proportion of beads which does not have this characteristic (*q* = 1 − *p*), *K* is the confidence level, and *E* is the desired sampling error.

Once the sample size was computed, the diameter (*d*) of the selected beads was measured with a digital Vernier. To determine the available surface (S_A_) to immobilize bacteria, the beads were considered as a sphere, and their surface was computed (S_A_ = π*d*^2^); after that, the surface per gram was obtained, considering the number of beads per gram. From this information, the best kind of beads was determined.

### 2.2. Bacteria Immobilization

Regarding the biofilm formation, 50 mL of agar (17 g of Cetrimide L^−1^ agar) previously inoculated with *P. koreensis* (24 h) were mixed with 5 g of calcium alginate beads. The mixture was incubated at 30 °C with orbital shaking at 150 rpm for 24 h. The biofilm formation was confirmed by scanning electron microscopy (SEM): the sample was prepared with a fixation with 2% formaldehyde and 70% ethanol for 18 h, subsequently dehydrated in ascending ethanol gradients: 70, 80, and 95%.

To find the best biosorption conditions, a statistical design of experiments was proposed considering the following variables: temperature (20 and 30 °C), amount of adsorbent (0.021, 0.024, and 0.027 g), and pH (adjusted to 6.6 and unadjusted). The adsorption tests were carried out in batch reactors stirred at 100 rpm, with a reaction volume of 30 mL and an initial metal concentration of 10 mg L^−1^; all tests were performed in triplicate.

### 2.3. Kinetics Study

The study was carried out using different Cr(VI) concentrations in the range of 10 and 120 mg L^−1^. For the preparation of the aqueous Cr(VI) stock solution, a specific amount of anhydrous K_2_Cr_2_O_7_ (2.828 g) was dissolved in distilled water; the volume of the solution was made up to 1 L to obtain a concentration of 1000 mg L^−1^. Solutions of different Cr(VI) concentrations, i.e., 10, 20, 30, 50, and 100 mg L^−1^, were prepared by dilution from the stock solution. After that, a concentration of 0.36 g L^−1^ of adsorbent was placed with 100 mL of Cr(VI) solutions at different initial concentrations; the mixture was kept in orbital shaking considering 100 rpm for 36 h at 30 °C. All experiments were carried out by duplicate.

The amount of Cr(VI) adsorbed by the immobilized bacteria was calculated using the following equation:(2)$$q=\frac{\left(C_{i}-C_{f}\right)V}{m}$$
where *q* (mg_metal_ mg_adsorbent_^−1^) represents the uptake of the metal by the biofilter, *V* is the reaction volume (L), *C_i_* is the initial concentration of Cr (mg L^−1^), *C_f_* is the final concentration of the metal (mg L^−1^), and m is the mass of the bioadsorbent on a dry basis (g).

The obtained data from the experiments were used to build the adsorption isotherms, which are analyzed considering the models of Langmuir and Freundlich, which are presented in Equations (3) and (4), respectively.
(3)$$q_{e}=\frac{q_{max}K_{L}C_{e}}{1+K_{L}C_{e}}$$
(4)$$q_{e}=K_{F}C_{e}^{1/n}$$
where *C_e_* is the adsorbate concentration in the solution at equilibrium, *q_max_* is the maximum adsorption capacity of the bioadsorbent, *q_e_* is the amount of adsorbate adsorbed per unit weight of adsorbent at equilibrium, *K_L_* is the Langmuir equilibrium constant, *K_F_* is the Freundlich constant, and 1/*n* is defined as the adsorption intensity according to with the next criteria: for (1/*n*) = 1, the relative adsorption (adsorption partition) is a linear function of the adsorbate concentration; when 0.7 < (1/*n*) < 1, the relative adsorption decreases as the concentration increases, which implies saturation of adsorption sites; (1/*n*) < 0.7 describes highly curved isotherms; finally, cooperative adsorption is represented by (1/*n*) > 1. Regarding the Langmuir model, the separation factor *R_L_*, also known as the equilibrium parameter, is computed with the following equation:(5)$$R_{L}=\frac{1}{1+K_{L}C_{i}}$$

The separation factor can be used as a guide to determine the adsorption strong or adsorbent/adsorbate interaction according to the following criteria: *R_L_* > 1 implies non-favorable conditions for adsorption, *R_L_* = 1 corresponds to a linear adsorption function, *R_L_* = 0 represents non-reversible adsorption and 0 < *R_L_* < 1 implies favorable conditions for adsorption [31].

To ease the computation of the parameters *q_max_*, *K_L_*, a reciprocal linearization of Equation (3) is recommended; for the computation of the parameters 1/*n* and *K_F_*, a logarithmic linearization of Equation (4) is suggested. The linearized equations are (6) and (7), respectively.
(6)$$\frac{1}{q_{e}}=\frac{1}{q_{max}}+\frac{1}{q_{max}K_{L}C_{e}}$$
(7)$$\log q_{e}=\log K_{f}+\frac{1}{n}\log C_{e}$$

To evaluate the rate of the adsorption, pseudo-first and pseudo-second order kinetic models were applied [19,32,33]. The linear form of those models is represented by Equations (8) and (9), respectively.
(8)$$\log\left(q_{e}-q_{t}\right)=\log(q_{e})-\frac{K_{1}}{2.303}t$$
(9)$$\frac{t}{q_{t}}=\frac{1}{K_{2}\cdot q_{e}^{2}}+\frac{t}{q_{e}}$$
where *q_e_*, *q_t_*, *K*_1_, and *K*_2_ are the equilibrium adsorption capacity, adsorption capacity at time *t*, first-order constant, and second-order constant, respectively. For the pseudo-first-order equation, the plot of left side versus time provides a linear graph from a linear regression; from the slope of this graph, the constant *K*_1_ is computed. Regarding the pseudo-second-order model, the intercept and slope of the line obtained from the linear regression of *t*/*q_t_* versus time are used to obtain *K*_2_.

### 2.4. Determination of Dry Weight of Bacteria Attached to Alginate Beads

The amount of adsorbent (*P. koreensis*) in the system was determined with the following procedure: 50 g (by wet weight) of the biofilter (*P. koreensis* adhered to the alginate beads) were taken and washed three times; the biofilm was removed by adding 200 mL of Tween 80 (0.1%) to the beads and stirring for 3 h in a magnetic stirrer [34]. Subsequently, the moisture on the supernatant obtained from the stirring stage was removed by drying the material in an oven at 110 °C until a constant weight was achieved.

## 3. Results

### 3.1. Alginate Beads

The characteristics of the beads obtained using the different micropipette tips are listed in Table 1. The size lends noticeable differences to the beads, especially for the amount and the available surface to carry the microorganisms. This inference is corroborated by the ANOVA (Table 2). Then, since F > FCV, significant differences exist between the three synthesized beads’ sizes. The relationship between the beads’ size and the surface per gram was considered to determine the best bead size: the fewer the beads, the larger the available surface to immobilize *P. koreensis*. Then, beads with S = 12.56 mm^2^ (2 mm diameter) were allowed to obtain the largest contact surface for the biofilter, and it was selected for successive experiments.

### 3.2. Biosorption System Implementation

The adhesion of the bacteria to the alginate beads was verified by observation under a scanning electron microscope (SEM at 5000×), complemented by an energy-dispersive X-ray spectroscopy (EDS) analysis (XL-30 ESEM, Philips; North Billerica, MA, USA). First, virgin alginate beads were analyzed; the results are shown in Figure 1.

It can be appreciated there is a coarse surface in the micrograph of Figure 1, which corresponds to virgin alginate beads. The EDS analysis showed that Ca is the main component of the beads, and no other elements are present since the beads were not in contact with either the bacteria or other material; this agrees with the report of Ohemeng-Boahen et al. [35].

After that, representative samples of beads taken from the immobilization system were analyzed at 8, 12, and 24 h. The respective micrographs are presented in Figure 2, where the adhesion of *P. koreensis* to the support is demonstrated.

At 8 h (Figure 2a), the micrograph is already different from the obtained for the virgin bead. It is possible to observe the presence of bacteria colonies randomly dispersed in space, forming a biofilm over the surface of alginate beads. The micrograph analysis also reveals the presence of cell aggregates, gaps, and embedded biomass in the alginate matrix, approximately 1 µm in length, which is consistent with the characteristics *P. koreensis*.

The micrograph of the beads sampled at 12, and 24 h (Figure 2c,e) shows slight changes compared with that obtained for the beads sampled at 8 h. The bacillary forms of *P. koreensis* are clearly remarked on the surface of the alginate beads. Then, it is deduced that 8 h is enough time for biofilm formation.

The presence of *P. koreensis* on the surface of the beads was confirmed by EDS analysis. The characteristic elements of the microorganisms’ prokaryotic wall (P, N, K) were identified in the spectrums shown in Figure 2b,d,f, which are similar to those found for other *Pseudomonas* species [36,37]. None of those above characteristics was observed in the virgin beads.

### 3.3. Cr(VI) Removal Tests for Optimization

The data obtained from the batch tests described in Section 2.2 were processed by an ANOVA at 95% confidence to determine the best operating conditions. Table 3 presents the values of the corresponding analysis, considering the associated variability of time and dose of bio-adsorbent and their interactions.

It should be noted that significant differences were obtained in the triple interaction pH, temperature, and adsorbent concentration. This is because the effect of a factor is defined as the change in the response produced by a difference in the factor level; in this case, almost all the terms were statistically significant (*p* < 0.05), except the influence of temperature and the interaction of the temperature versus the concentration of the adsorbent.

However, even if this analysis allows determining that the experiments show statistically significant differences, it is not possible to obtain information regarding which treatment is different from the others. Then, a Fisher test was performed (Table 4). This test allows direct comparisons between different treatments of individual groups, through a less significant difference test [38].

Thus, from the results of the Fisher test, it is concluded that Treatment 1 (30 °C, pH 6.6, amount of adsorbent 0.027 g) was the best one. Therefore, these conditions were selected for the subsequent experiments. It is important to remark that the selected conditions agree with the optimal growth conditions of *Pseudomonas* [39]. On the other hand, the quantity of microorganisms in a biosorption system is directly proportional to its efficiency; this situation confirms that the best treatment has the largest number of microorganisms.

Although the results of Fisher’s test indicate that Treatment 1 corresponds to the best adsorption conditions, other treatments allow reaching similar final chromium concentrations. Therefore, it is inferred that *P. koreensis* present biological activity at 20 and 30 °C, and the two pH tested, thus proving to be a robust microorganism. This represents an advantage for continuous reactors and scaling up the adsorption system.

### 3.4. Biosorption Effect on the Biofilter Surface

Samples of the biofilter were taken before adsorption (0 h) and after Cr adsorption (48 h). The samples were analyzed by SEM and EDS to determine morphological and chemical changes that occurred on the biosorbent surface due to the metal adsorption process, and to visualize the interaction between the adsorbed contaminants and the *P. koreensis*/alginate beads system. The corresponding results are presented in Figure 3.

At 0 h, the biofilm on the surface is still clearly observed (Figure 3a) as for the biofilter preparation stage (Section 3.2); in addition, the EDS analysis (Figure 3b) allows corroboration of the presence of *P. koreensis* on the alginate beads through the signals of P, N, and K, which conform to the cell wall.

On the other side, at 48 h of experimentation, the corresponding micrograph (Figure 3c) shows that *P. koreensis* are still on the alginate beads after the bioabsorption process. However, from the EDS analysis, it is not possible to demonstrate the presence of the characteristic elements of the bacterial cell wall (Figure 3d) as for the previous analysis. This may be explained since, although the bacteria are still present in the biofilter, they are highly dispersed.

For this reason, it was necessary to resort to another analytical technique to better understand how the process of adsorption of contaminants was carried out on the *P. koreensis*/alginate system. Fourier transform infrared (FTIR) analysis was selected for this goal for two reasons: it is possible to identify functional groups, and the preparation of the sample to be analyzed allows including a larger number of bacteria, so then the microorganisms can be analyzed with greater precision. The FTIR spectrum of samples taken before (0 h) and after (32 and 48 h) the adsorption is presented in Figure 4 and Figure 5, respectively.

The FTIR spectrum before the adsorption (Figure 4) exhibits several bands corresponding to the characteristic functional groups of the bacteria biochemical compounds, specifically peptidoglycans, proteins, and lipopolysaccharides [40]. The signal centered at 3293 cm^−1^ is associated with the vibration of hydroxyl and amino groups (O-H and N-H stretching) corresponding to peptidoglycans [41]. Likewise, a characteristic band of C-H is detected at 2929 cm^−1^; the signal of the carboxylic group C=O is observed around 1626 cm^−1^, and the symmetrical stretching of C-O-C at 1038 cm^−1^. Finally, the functional groups of nitrogen of the amino groups can be observed at 1425 cm^−1^, which corresponds to the C-N bond, as well as the vibration outside the plane of the group N-H at 929 cm^−1^ [42]. The FTIR spectrum seems like the one obtained for *Arthrobacter viscosus* [41] and for *Pseudomonas* species such as *P. alcaliphila* [37] and for *P. stutzeri* [43].

The FTIR spectrum of the biofilter samples taken after the adsorption (Figure 5) showed some changes from the spectrum of the sample taken before the adsorption. First, for the spectrum of the sample taken at 32 h, an intensity decrease of the band centered at 3293 cm^−1^ (assigned an O-H stretching vibration) was observed (Figure 5a). Meanwhile, the bands related to C-H bonds (~2900 cm^−1^), showed no remarkable changes. On the other hand, a remarkable change in the carbonyl group C=O band was observed; the intensity of this band is drastically reduced in the sample taken at 32 h (Figure 5a) until it almost disappears in the sample taken at 48 h (Figure 5b). Moreover, a shift in C-O-C symmetric stretch from 1038 to 1090 cm^−1^ was observed. Finally, a relative increase in the intensity of the bands corresponding to the N-H bond was observed at 32 and 48 h after adsorption. These remarks allow us to deduce that the functional groups O-H, and C=O are the main ones responsible for the adsorption of Cr(VI) in this system. This behavior is similar to that reported for *A. viscosus* in a Cr(VI) adsorption batch reactor [41]; the authors associated the removal of the metal to the carboxyl, phosphoric, or amine groups. In addition, the FTIR spectrum after adsorption presented some differences in comparison with those observed by El-Naggar et al. [37] for *P. alcaliphila* removing Cr(VI); due to the modifications in the bands of the spectra, the authors associate the Cr(VI) adsorption to groups such as phenolic, carbonyl ester, acetyl, carboxylate, alkanes, and carbonyl. These differences could be due to the specific characteristics and metabolism of the *Pseudomonas* genus.

### 3.5. Kinetic Adsorption Experiments

Here, it is important to mention that in preliminary assays it was observed that the pristine alginate beads have a considerable contribution in the chromium removal (about 20%); however, the *P. koreensis* presence in the support increased the removal until it reached nearly 100% in a synergic process. Then, considering the results of the tests for optimization (Section 3.3), a new experimental stage was implemented: a series of adsorption kinetic assays were performed to identify interactions in the adsorbent/adsorbate system. The operating conditions considered were: pH 6.6, stirring 100 rpm, temperature 30 °C, initial concentrations of Cr(VI) of 0, 10, 25, 40, 55, 70, 85, 100, and 120 ppm, amount of adsorbent (bacteria dry weight) remained constant as 0.36 g L^−1^, and reaction volume (Cr(VI) solution) was 100 mL. The experiments were performed by duplicate, and samples for monitoring were taken at 0, 4, 8, 12, 24, 28, 30, 32, 34, and 36 h. The adsorption system behavior is shown in Figure 6.

The adsorption is progressively developed following a similar trend with all initial concentrations assayed; at 30 h, a quasi-equilibrium point is reached in all cases. After that time, lower concentrations remain in the equilibrium, but higher concentrations still present increasing adsorption. This implies that the immobilized bacteria can remove even larger concentrations of Cr. Since the experimentation time was restricted to 36 h, and knowing that the adsorption achieved at higher concentrations could be larger, the quasi-equilibrium point at 30 h was taken for further analysis.

It is pertinent to indicate that none of the experiments demonstrated the formation of a precipitate at the bottom of the reactor, which indicates that during the chromium adsorption process, there were no reactions for the formation of compounds not assimilable by the biofilter.

Further analysis of the adsorption isotherms using the Langmuir and Freundlich models was performed to determine the strength and type of interaction between the biofilter and the contaminant. Even if these models were proposed to describe the adsorption of gases on a solid, they are used for other systems, since it is accepted that they provide valid information for more complex systems such as biological ones, as reported in several studies (Table 5).

Then, for the isotherm analysis, it was considered that the biofilter saturation was reached at 30 h. Figure 7 includes the linearized isotherms; Table 6 lists the corresponding parameters obtained from Equations (5) and (6) and the linear equation representing the isotherm.

The linearization of the adsorption isotherms data in the Langmuir model coordinates is shown in Figure 7a; a determination coefficient R^2^ = 0.981 was obtained, indicating a good fitting of the adsorption systems by this model.

Concerning the adsorption rate, Table 7 lists the correlation coefficients and constants for low, middle, and high initial concentrations.

## 4. Discussion

The coarse morphology observed on virgin alginate beads agrees with those reported in other works [49]; this characteristic seems adequate for a bacteria immobilization as reported for other materials used as microorganism carriers [37,50,51].

The biofilm formation on the surface of alginate beads is similar to that reported by Mangwani et al. [52], who observed *P. mendocina* on the glass; also, El-Naggar et al. [37], achieved the immobilization of *P. alcaliphila* in sodium alginate beads. In their studies, using SEM, the authors verified the presence of a biofilm composed of bacillary aggregates [37,52].

The maximal adsorption capacity obtained from experiments (q_max-ex_) is 168 mg g^−1^; this value is similar and even better than the results obtained by other authors in several heavy metals biosorption systems [14,19,22,26]. In this sense, the *P. koreensis*/alginate beads biofilter tested in the present study is a promising option for the decontamination of water containing Cr(VI). In addition, from the kinetic study and using the Langmuir adsorption isotherm, it was determined that the theoretical *q_max_* is 625 mg g^−1^; the values is high in comparison with other works. In this context, further studies could be developed to prove this theoretical *q_max_* by experiments.

From the application of the Langmuir model to the *P. koreensis* kinetics, a separation factor (*R_L_* = 0.6304) was obtained that satisfies the condition 0 < *R_L_* < 1, which corresponds to a favorable adsorption process. That means the *P. koreensis*/alginate beads system presents adsorption characteristics to attract and to grab the metal [13,31]. Moreover, the *R_L_* value is in the middle of the interval |0 1|, which is interpreted as the limits for reversibility since *R_L_* = 0 corresponds to irreversible adsorption and *R_L_* = 1 to reversible adsorption [13,33]; then, it is considered that the strength of the adsorption by *P. koreensis* is enough to remove Cr(VI) and it is also convenient for desorption, which is useful for biofilter regeneration.

Regarding the Freundlich model, the behavior of the Cr(VI) adsorption indicates a directly proportional relationship between the adsorption capacity of the adsorbent material (*P. koreensis*/alginate system) and the chromium concentration at the equilibrium state. A quite acceptable correlation was obtained since the coefficient of determination is R^2^ = 0.978. According to the fundamentals of the Freundlich model [53], the behavior of the Cr(VI) adsorption indicates a directly proportional relationship between the adsorption capacity of the adsorbent material (*P. koreensis*/alginate beads system) and the chromium concentration at the equilibrium state. Moreover, the parameter 1/*n* < 1 implies the adsorbent material is able for normal adsorption, which corresponds to the separation factor obtained from the Langmuir model. In addition, since 1/*n* = 0.9345 matches the condition, 0.7 < (1/*n*) < 1, it is inferred that the relative adsorption of Cr(VI) by the *P. koreensis* biofilter decreases as the concentration of the metal increases, which implies that saturation of adsorption sites is occurring, as it was deduced for another biological adsorption system [54].

In addition, the determination coefficient for both models is similar (R^2^ = 0.978 for Freundlich and R^2^ = 0.981 for Langmuir), which implies that the two approaches describe the adsorption process. A similar situation (adjustment to both models Freundlich and Langmuir) has been observed in other biosorption systems [41,48,54]. For Cr(VI) removal, some authors proposed that the behavior of adsorption in microorganism/support systems is as a monolayer with heterogeneous distribution of active sites [32], and including intracellular bioaccumulation [41]. This could suggest that the *P. koreensis*/alginated beads system forms a biofilm with a homogeneous distribution of active sites in the surface where Cr(VI) is adsorbed. After that, *P. koreensis* also assimilates the metal inside its cellular structure; then, the adsorption system follows a multilayer adsorption. For this reason, the capacity for chromium removal is high. This highlights the potential of *P. koreensis* to remove heavy metals, and that it is suitable for further investigations with other operating conditions, higher adsorption volumes, and reactor configurations.

Moreover, the experimental data better fit the pseudo-second-order model, which corresponds with the reported in other sorption studies [19]. Therefore, it can be deduced that the adsorption depends on the active sites available on the *P. koreensis*/alginate beads system, which acts as a biosorbent [19,33]. In addition, this situation suggests that adsorption of Cr(VI) on the biofilter occurs, which corresponds to the obtained from the adsorption isotherms [32].

On the other hand, FTIR analysis suggests a bonding between the hydroxyl groups of the biofilter surface and the chromium when the signal at ~3300 cm^−1^ after the adsorption process was reduced in intensity. Moreover, since the signals observed at ~2920 cm^−1^ did not suffer notorious changes before and after adsorption, the reduction of Cr(VI) to Cr(III) [55] was discarded.

## 5. Conclusions 

The biofilter formed by *P. koreensis*/alginate beads is able to remove up to 96% of Cr(VI) in an aqueous solution at 30 °C and pH 6.6. A theoretical maximal adsorption capacity of *q_max_* = 625 mg g^−1^ was determined. From the isotherm analysis, it was observed that both models, Langmuir and Freundlich, fit the experimental data. This suggests that the adsorption is performed in homogeneous distribution of active sites and multilayer adsorption, since *P. koreensis* also assimilate chromium in their cellular structure, according to the separation parameter. In addition, the functional groups O-H and C=O, which are part of the prokaryotic wall of bacteria, were identified as responsible for the adsorption of Cr(VI).

## 6. Perspectives

From this research, it is possible to mention that *P. koreensis* could be used as a basis to develop systems for water purification. However, complementary works are required to evaluate the *P. koreensis* performance to remove heavy metals in presence of other elements such as anions, cations and multimetals, as well as in real water environments. Moreover, the recycling of the proposed biofilter should be considered for further studies. It is required to determine whether alginate beads are able to keep the mechanical resistance for several cycles; also, the system *P. koreensis*/alginate beads should be evaluated economically for scaled-up application.

On the other hand, a deeper analysis has to be conducted to solve the questions related to the nature of the reactions occurring during the adsorption process. In fact, this kind of study could help to understand the mechanisms better and eventually predict a specific behavior with similar systems or parameter change.

However, once the biofilter is exhausted and it is impossible to regenerate, a disposal strategy is required. Underground enclosure is an easy method to dispose of the materials; but other options should be explored. In this context, circular economy approaches can be an alternative: the adsorbed metal could be recovered through thermal treatment of the biofilter; after that, the recovered metal could be used in its current applications, obtaining economic benefits.

## Figures and Tables

**Figure 1 ijerph-20-01385-f001:**
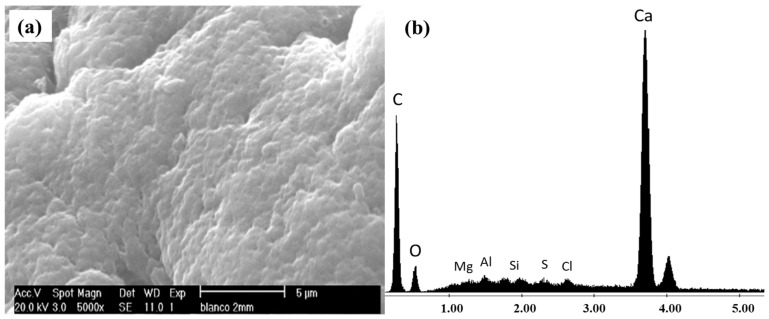
(**a**) SEM and (**b**) EDS analysis of virgin alginate beads.

**Figure 2 ijerph-20-01385-f002:**
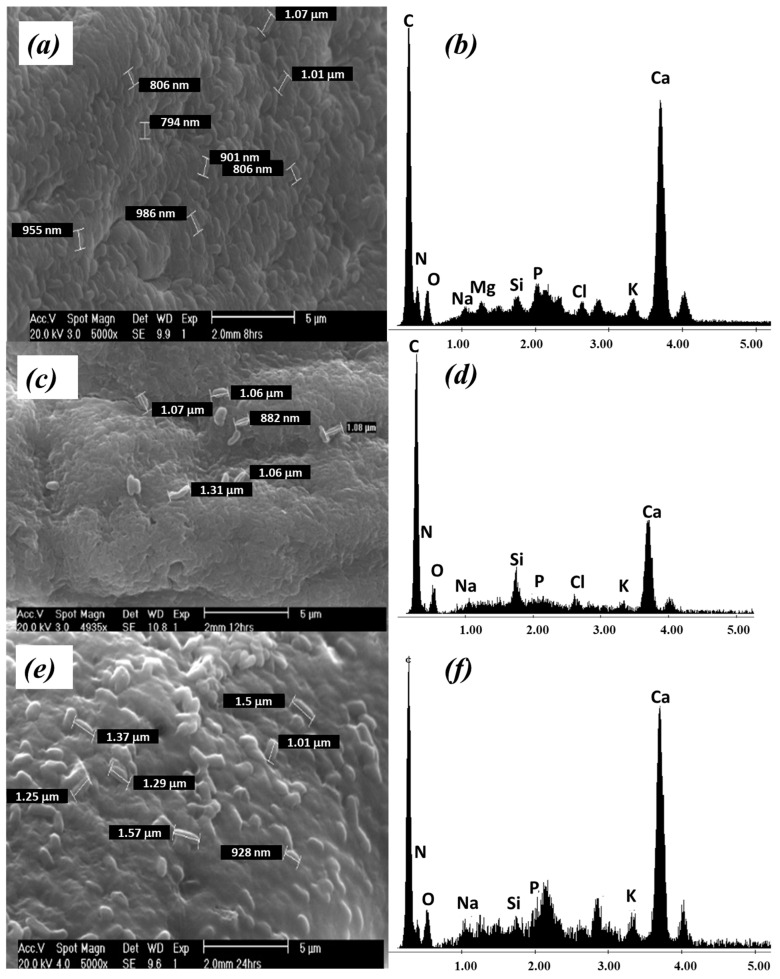
*P. koreensis* supported on alginate beads at different time: (**a**) MEB and (**b**) EDS at 8 h; (**c**) MEB and (**d**) EDS at 12 h; (**e**) MEB and (**f**) EDS at 24 h.

**Figure 3 ijerph-20-01385-f003:**
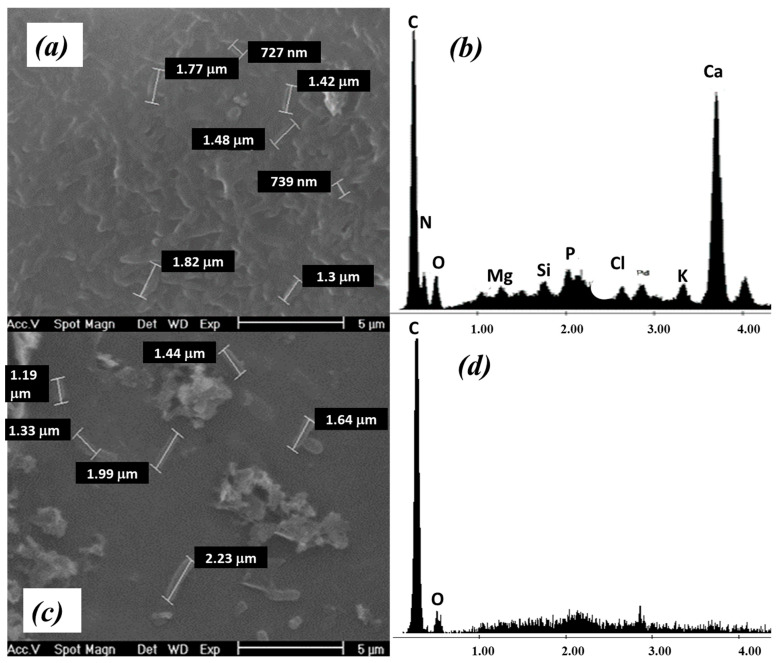
SEM and EDS analysis of *P. koreensis*/alginate beads system after Cr adsorption at (**a**,**b**) 0 h and (**c**,**d**) 48 h.

**Figure 4 ijerph-20-01385-f004:**
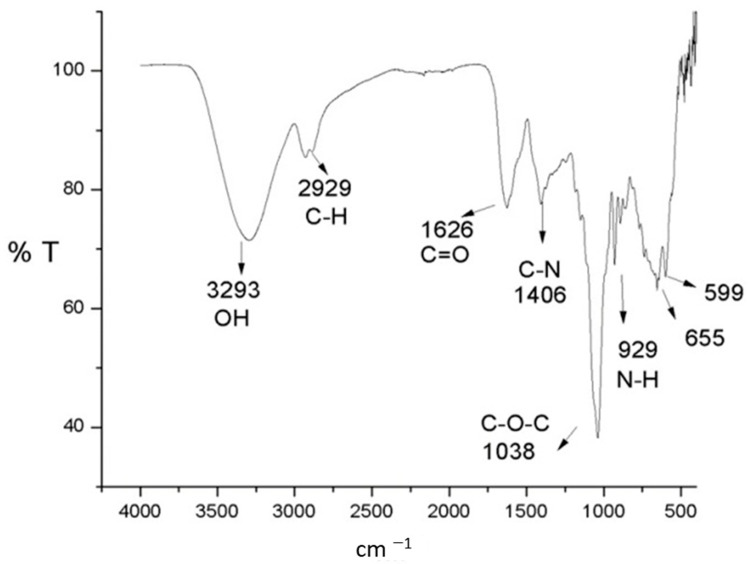
FTIR spectrum of the *P. koreensis*/alginate beads system at 0 h.

**Figure 5 ijerph-20-01385-f005:**
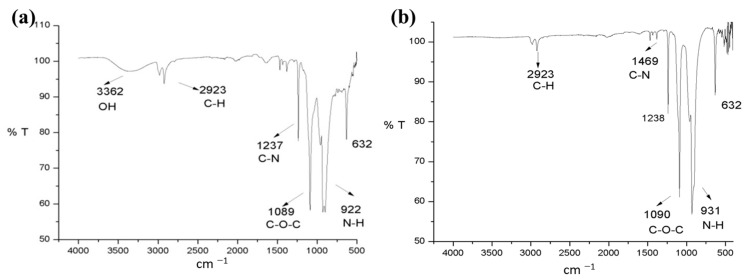
FTIR spectra of the *P. koreensis*/alginate beads system after the adsorption at (**a**) 32 h and (**b**) 48 h.

**Figure 6 ijerph-20-01385-f006:**
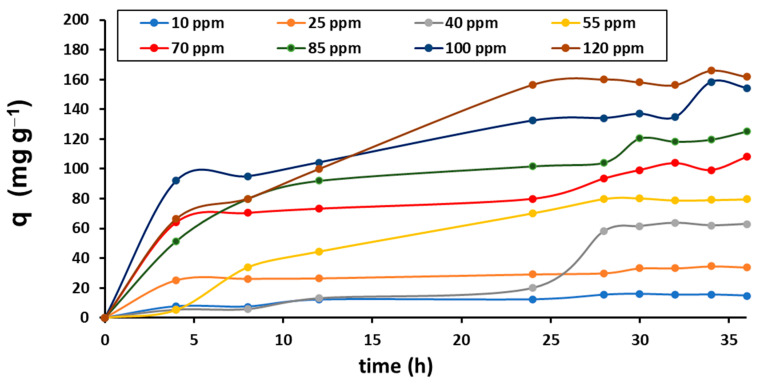
Cr(VI) adsorption kinetics.

**Figure 7 ijerph-20-01385-f007:**
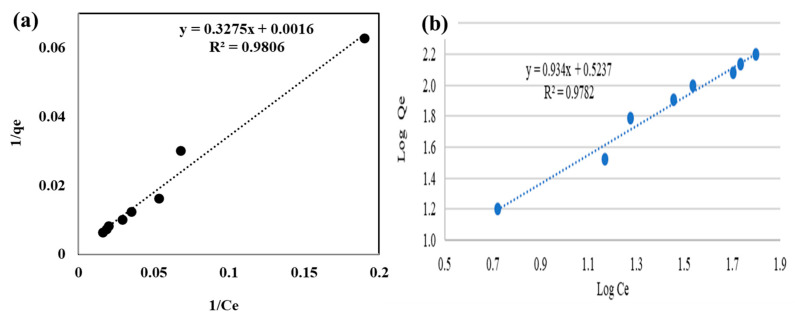
(**a**) Reciprocal linearization, and (**b**) logarithmic linearization of the Cr(VI) adsorption isotherm.

**Table 1 ijerph-20-01385-t001:** Alginate beads prospects to immobilize *P. koreensis*.

Tip	Diameter nm	Items Beads/g	Set Weight g	Total Items Beads/Set	Surface mm^2^/Bead	Surface mm^2^/g
blue	3.0	35.33	65	2296	28.27	998.9
yellow	2.4	84.30	56	4721	18.09	1525.6
white	2.0	130.66	60	7839	12.56	1641.2

**Table 2 ijerph-20-01385-t002:** Results of ANOVA for the size of alginate beads.

	Sum of Squares	Degree of Freedom	Mean Square	F	*p*	F_CV_
Between groups	13,643.56	2	6821.78	2558.17	1.6 × 10^−9^	5.14
Within groups	16	6	2.67			
Total	13,659.56	8				

**Table 3 ijerph-20-01385-t003:** ANOVA for the optimization of the best biosorption conditions.

Effect	Sum of Squares	Degree of Freedom	Sum of Means	F	*p*
Interception	3332.79	1	3332.79	85,506.38	0.000
Temperature	0.05	1	0.05	1.35	0.254
pH	9.08	1	9.08	232.97	0.000
[Adsorbent]	66.16	3	22.05	565.77	0.000
Temperature × pH	0.59	1	0.59	15.12	0.001
Temperature × [Adsorbent]	0.13	3	0.04	1.14	0.348
pH × [Adsorbent]	6.22	3	2.07	53.19	0.000
Temperature × pH × [Adsorbent]	038	3	0.13	3.24	0.035
Error	1.25	32	0.04		

**Table 4 ijerph-20-01385-t004:** Fisher test to find the best biosorption conditions.

T (°C)	pH	Beads (gr)	Cr(VI) (mg L^−1^)	Treatment			
				1	2	3	4
30	6.6	0.027	6.41	*			
20	6.6	0.027	6.95		*		
30	6.6	0.024	7.06		*	*	
20	6.6	0.024	7.17		*	*	
30	6.6	0.021	7.37			*	*
20	6.6	0.021	7.52				*

* indicates statistically similar result for the tested conditions.

**Table 5 ijerph-20-01385-t005:** Remotion of Cr(VI) by different bacteria.

Bacteria	Support	Model	R^2^	*q_max_* (mg g^−1^)	References
*A. viscosus*	None	Langmuir	0.96	1161	[41]
*Staphylococcus epidermidis*	Kaolin	Langmuir Freundlich Temkin Dubinin-Radushkevich	0.99 0.99 0.80 0.82	56	[32]
*Escherichia coli*	Kaolin	Langmuir Freundlich Temkin Dubinin-Radushkevich	0.99 0.99 0.82 0.79	91	[32]
*Escherichia coli*	Waste tea biomass	Langmuir Freundlich Temkin	0.99 0.91 0.92	16.75	[44]
*Ochrobactrum intermedium*	None	Langmuir	0.96	51.96	[45]
*Cupriavidus metallidurans*	None	Langmuir	0.90	32.63	[45]
*Mycobacterium* sp.	None	Langmuir Freundlich Temkin Hill-der Boer D-R	0.99 0.92 0.97 0.98 0.97	61.51	[46]
*Cronobacter muytjensii*	None	Langmuir	-	73.8	[47]
*P. alcaliphila*	Alginate beads	Langmuir Freundlich	0.99 0.92	10	[37]
*P. koreensis*	Alginate beads	Langmuir Freundlich	0.98 0.97	625	This study
*P. stutzeri*	None	Langmuir Freundlich Temkin Dubinin-Radushkevich	0.89 0.99 0.98 0.71	27.47	[43]
*Pseudomonas* sp.	None	Langmuir Freundlich	0.95 0.99	95	[48]

**Table 6 ijerph-20-01385-t006:** Langmuir and Freundlich parameters for Cr(VI) adsorption.

Model	*K_L_*	*q_max_* (mg g^−1^)	*R_L_*	*K_F_*	1/*n*
Langmuir	4.88 × 10^−3^	625	0.6304	-	-
Freundlich	-	-	-	3.33	0.9345

**Table 7 ijerph-20-01385-t007:** Pseudo-first- and pseudo-second-order parameters.

Model	Parameters	10 ppm	70 ppm	120 ppm
Pseudo-first-order	*K* _1_	0.101	0.075	0.179
	R^2^	0.671	0.818	0.934
Pseudo-second-order	*K* _2_	0.002	0.002	0.0003
	R^2^	0.777	0.966	0.961

## Data Availability

Not applicable.

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
