# Peer review of "High Adsorption of Hazardous Cr(VI) from Water Using a Biofilter Composed of Native Pseudomonas koreensis on Alginate Beads"

_ijerph, 2023, doi:10.3390/ijerph20021385_

Round 1

Reviewer 1 Report

The authors have conducted a very interesting research. They used the biofilter formed by P. koreensis/alginate beads for removal Cr(VI) from aqueous solutions.

Regardless of the clear mistakes that crept in during the writing of the manuscript, unfortunately, the manuscript left too many unresolved issues related to the obtained results, which should be elaborated in more details.

The FTIR analysis is questionable. The authors explained the FTIR spectra by supporting the obtained results with works of others who explained the bands, and they also obtained those bands, but they ignored them. Some bands that they claim that have disappeared are still present in the spectra.

Also, kinetic and equilibrium data are also questionable. The authors stated that the kinetics study was performed for the 24 hours and the results presented are for 36 h. Nevertheless, the authors presented the SEM images and FTIR spectra after 48 hours.

As for the kinetic data, at lower concentrations the equilibrium is reached, but at higher concentrations the establishment of equilibrium is questionable. More samples had to be taken to make sure that the equilibrium is reached. With such unreliable data, the authors tested the data with two adsorption isotherm models which is not recommended. Moreover, the authors took the value of qmax obtained by the model as the relevant value, and not the one obtained by the experiment.

Unfortunately, considering the above, this is not a well written manuscript and I must reject it.

Author Response

High adsorption of hazardous Cr(VI) from water using a biofilter composed of native Pseudomonas koreensis on alginate beads

Manuscript Id: ijerph-2070512

ANSWERS TO REVIEWER 1

Authors thank the reviewer for his remarks. We provide in this document answers for each comment. We hope this information helps to improve the manuscript.

Reviewer 2 Report

This study synthesized the biofilter by immobilizing a native Pseudomonas koreensis on calcium alginate beads. The biofilter exhibited an excellent removal efficiency of Cr(VI) in aqueous solutions at 30 °C and pH of 6.6. This finding provided a potential application of P. koreensis in water treatment. The experimental work was conducted rigorously and the manuscript was well in written. However, some critical issues should be concerned:

1. The authors should add more information related to concentration level of chromium within different environmental matrices and provide a clear research scenario to further clarify the purpose of the study, such as natural water bodies, sewage treatment plants, etc.

2. The adsorption mechanisms was not clearly presented: 1) How to quantify the contribution of Cr(VI) adsorbed on virgin alginate beads ? 2) In addition to the functional groups, whether the specific areas, REDOX potential and etc of biofilter are involved in sorption process? 3) Did redox reactions, chelation, and precipitation occur during adsorption process?

3. The authors should add and clarify the impact of water constituents including anions, cation and dissolved organic matter to Cr(VI) sorption. In addition to laboratory simulation, the authors need to supplement the effects of biofilter on Cr(VI) sorption in real water bodies (freshwater, marine groundwater or sewage treatment plant influent, etc.)  

4. In view of the practicality and recycling availability of biofilter, the authors should add changes of Cr(VI) sorption performance after multiple recycling of the biofilter.

5. Lines 371-373, it is unreasonable to judge the changes of adsorption based on 0.7<(1/n)<1.

6. Lines 370 and 380, “a homogeneity in the bioactive sites on the surface of the adsorbent” and “a biofilm with a heterogeneous surface where Cr(VI) is adsorbed” is contradictory.

7. Some spelling mistakes should be corrected: line 36, change “organisms” to “organizations”; line 107 change “K the confidence level, and E the desired sampling error” to “K is the confidence level, and E is the desired sampling error”.

8. Please unify the writing form of carbonyl and infrared spectra in this manuscript: CO= (line 21), C=O (line 282), O=C (line 296), O=C- (line 393), Infrared Analysis (IR) (line 270), IR spectrum (line 277) FTIR spectra (line 285) and etc.

9. In figure 3, where is figure 3e?

Author Response

High adsorption of hazardous Cr(VI) from water using a biofilter composed of native Pseudomonas koreensis on alginate beads

Manuscript Id: ijerph-2070512

ANSWERS TO REVIEWER 2

Authors thank the reviewer for his remarks. We provide in this document answers for each comment. We hope this information helps to improve the manscript.

Reviewer 3 Report

The authors have synthesized a biofilter for water purification from CrVI. The filter is made of Pseudomonas koreensis on calcium alginate. The synthesis and characterization of the material before and after adsorption is well presented. The obtained results are significant and should be published with minor corrections:

-          In the experimental part, write how the initial solutions for determining the adsorption capacity were made.

            -           If it is possible to determine the order of the reaction.

In order for the work to have greater significance, it is SUGGESTED to supplement the conclusion. For example:

-          Add "5. Conclusions and future perspective"

     -      Future perspectives should be related to the accumulation of used biofilters containing CrVI. If possible, propose a solution to the possible use of saturated filters, as a problem that needs to be solved.

Author Response

High adsorption of hazardous Cr(VI) from water using a biofilter composed of native Pseudomonas koreensis on alginate beads

Manuscript Id: ijerph-2070512

ANSWERS TO REVIEWER 3

Authors thank the reviewer for his remarks. We provide in this document answers for each comment. We hope this information helps to improve the manuscript.

Round 2

Reviewer 1 Report

The authors have conducted a very interesting research. They used the biofilter formed by P. koreensis/alginate beads for removal Cr(VI) from aqueous solutions.

Although the Authors made some significant corrections, they are still not sufficient for the manuscript to be accepted.

The Authors explained that at the higher concentrations a quasi-equilibrium state was reached, but this is just an assumption. The maximal adsorption capacity obtained from experiments (q max-ex ) is 168 mg g-1. They stated in lines 423-426, page 4 that this capacity could be much larger and proved with further studies. There is no need for further studies, only a need to expand the experiment at higher concentrations until the equilibrium is reached.

It is unacceptable to fit the experimental data with the models from an unfinished experiment…

Unfortunately, considering the above, I must reject the manuscript once again.

Author Response

We thank again the reviewer for his comments. We provide in the corresponding answers in this document.

Comment: The authors have conducted a very interesting research. They used the biofilter formed by P. koreensis/alginate beads for removal Cr(VI) from aqueous solutions.

Answer: We thank the reviewer for the comment.

Comment: Although the Authors made some significant corrections, they are still not sufficient for the manuscript to be accepted.

Answer: We try to explain here our reasons.

Comment: The Authors explained that at the higher concentrations a quasi-equilibrium state was reached, but this is just an assumption. The maximal adsorption capacity obtained from experiments (q max-ex ) is 168 mg g-1. They stated in lines 423-426, page 4 that this capacity could be much larger and proved with further studies. There is no need for further studies, only a need to expand the experiment at higher concentrations until the equilibrium is reached.

Answer: The experiments were planned with the concentrations reported in the manuscript; to extend the results, a new planning would be required, which should be performed in the frame of further studies. The results suggest the maximal adsorption capacity is larger than the obtained in the experiments; that is why a theoretical qmax is reported besides the experimental one. One of the advantages of using the models is that they help to deduce approximative behaviors without the need of excessive experiments, reducing time and resources.  The experimental qmax is a first result which is being considering as the basis for future research.

Comment: It is unacceptable to fit the experimental data with the models from an unfinished experiment.

Answer: The experiment was finished considering the planned conditions. The equilibrium was reached before around 30 h for lower concentrations; since these experiments were performed during 36 h, we decided to use that time for higher concentrations. It is important to remark that for higher concentrations, the samples were analyzed after the experiments were finished (due to the availability of the equipment in our institution). The graphics were obtained once the experiments were finished and we found the reported behavior. For this reason, we stated that further studies are required. 

Comment: Unfortunately, considering the above, I must reject the manuscript once again..

Answer: We thank the remarks of the reviewer; will take them in consideration in our future works.

Reviewer 2 Report

Manuscript ID: ijerph-2070512

Title: High adsorption of hazardous Cr(VI) from water using a biofilter composed of native Pseudomonas koreensis on alginate beads

 Comments:

Authors revised the original manuscript, however, some critical issues remain unresolved:

Comment 1: 1) whether the specific areas, REDOX potential and etc of biofilter are involved in sorption process? 2) Did redox reactions, chelation, and precipitation occur during adsorption process?

Comment: 3. The authors should add and clarify the impact of water constituents including anions, cation and dissolved organic matter to Cr(VI) sorption. In addition to laboratory simulation, the authors need to supplement the effects of biofilter on Cr(VI) sorption in real water bodies (freshwater, marine groundwater or sewage treatment plant influent, etc.).

Comment: 4. In view of the practicality and recycling availability of biofilter, the authors should add changes of Cr(VI) sorption performance after multiple recycling of the biofilter.

Author Response

Authors thank the reviewer for his remarks. We provide in this document answers for each comment. We hope this information helps to improve the manuscript.

Comment: Authors revised the original manuscript, however, some critical issues remain unresolved:

Comment 1: 1) whether the specific areas, REDOX potential and etc of biofilter are involved in sorption process?

2) Did redox reactions, chelation, and precipitation occur during adsorption process?

The authors should add more information related to concentration level of chromium within different environmental matrices and provide a clear research scenario to further clarify the purpose of the study, such as natural water bodies, sewage treatment plants, etc.

Answer: According to the FTIR spectrum, in the pristine biofilter, a signal was observed (~3300 cm-1) assigned to the vibration of hydroxyl groups, which we claim as the main adsorption sites for bonding chromium. After chromium adsorption, the FTIR spectrum showed a reduction in the signal at ~3300 cm-1, probably due to the bonding of the chromate ion on the biofilter surface. A reduction of Cr(VI) to Cr(III) was not proposed since no remarkable changes in signals at ~2900 and ~1730 cm-1, were observed 1. Concerning extracellular precipitation, it did not occur; at least it was not observed a precipitate at the reactor bottom along the adsorption assays. Complementary lines were added in the manuscript to mention these details (lines 450-478 and 561-565). Nevertheless, since we only measured the Cr(VI) concentration, we cant ensure the absence of Cr(III) species in the reaction process.

  • Nakano, K. Takeshita, T. Tsutsumi. Adsorption mechanism of hexavalent chromium by redox within condensed-tannin gel. Water Research, 2001, 35(2), 496-500.

Regarding the concentration level of chromium in environmental matrices, next sentences were included (Lines 38-43): “The maximal permissible level for chromium in water is 1 and 0.5 mg L-1 for agriculture and public uses, respectively. In soils, the maximum levels are 510 and 280 mg kg-1 for industrial and agriculture/commercial/residential uses, respectively [9]. Concentrations (µg L-1) of chromium in rivers and lakes around the world were determined as 388.77 (Africa), 383.93 (Asia), 13.61 (Europe), 5.42 (North America), and 903.78 (South America) [7].”

Comment: 3. The authors should add and clarify the impact of water constituents including anions, cation and dissolved organic matter to Cr(VI) sorption. In addition to laboratory simulation, the authors need to supplement the effects of biofilter on Cr(VI) sorption in real water bodies (freshwater, marine groundwater or sewage treatment plant influent, etc.).

Answer: This comment was addressed in lines 577-619 of the updated manuscript.

Comment: 4. In view of the practicality and recycling availability of biofilter, the authors should add changes of Cr(VI) sorption performance after multiple recycling of the biofilter.

Answer: This aspect is being considered in future work. The present report does not include the regeneration and recyclability of the biofilter; it just presents the potential of a native bacterium to remove chromium. In future work, chromium removal in a biofilter will be studied in a continuous system, with which the useful life of the biofilter and its regeneration capacity can be determined. This will make it possible to identify the biofilter efficiency in a scenario close to reality.